# Upper Limb’s Injuries in Agriculture: A Systematic Review

**DOI:** 10.3390/ijerph17124501

**Published:** 2020-06-23

**Authors:** Nicola Mucci, Veronica Traversini, Lucrezia G. Lulli, Antonio Baldassarre, Raymond P. Galea, Giulio Arcangeli

**Affiliations:** 1Department of Experimental and Clinical Medicine, University of Florence, 50139 Florence, Italy; nicola.mucci@unifi.it (N.M.); giulio.arcangeli@unifi.it (G.A.); 2Occupational Medicine School, University of Florence, 50139 Florence, Italy; lucreziaginevra.lulli@unifi.it; 3Doctoral School in Clinical Sciences, University of Florence, 50139 Florence, Italy; antonio.baldassarre@unifi.it; 4Faculty of Medicine & Surgery, University of Malta, Msida, MSD2090 L-Imsida, MSD, Malta; raymond.galea@um.edu.mt; 5The Malta Postgraduate Medical Training Programme, Mater Dei Hospital Msida, MSD2090 L-Imsida, MSD, Malta

**Keywords:** upper limb’s injuries, agriculture, farmworkers, health promotion, organizational behavioral measures, occupational medicine

## Abstract

Agriculture is one of the most hazardous economic sectors, and it accounts for many accidents and occupational diseases every year. In Italy, about one-third of injuries involve the upper extremity, with long-term consequences for the workers and economic damage for agricultural companies and farms. This systematic review describes upper limb injuries among farmworkers, especially hand injuries, and highlights the main dangerous risk factors. Literature review included articles published in the major databases (PubMed, Cochrane Library, Scopus), using a combination of some relevant keywords. This online search yielded 951 references; after selection, the authors analyzed 53 articles (3 narrative reviews and 50 original articles). From this analysis, it appears that younger male farmers are mostly involved, especially in the harvesting season. The upper limb and hand are often the body parts that sustain most damage as these are mostly involved in driving tractors or tools. The most frequent type of lesions are open wounds, lacerations, fractures, strains, and overexertion lesions. Sometimes, a distracting element (such as mobile phone use, quarrels, working hours load) is present; poor use of protective devices and lack of safety design in tools can also increase the risk of accidents. For these reasons, in the agricultural sector, a system of health promotion and good practices is needed to promote workers’ awareness of the sources of risk, highlight more dangerous situations and apply organizational behavioral measures.

## 1. Introduction

In Italy, about 1.1 million of workers are currently employed in the agricultural sector. According to recent statistics, Italian agricultural companies cultivating some kind of crops are 97.4% of the total, those specialized in permanent crops comprise 48.4%, with another 16% of all the companies having cattle farms. Most productive units are located in Puglia, Sicily, Calabria, and Campania, with over 700,000 agricultural activities (Figure 1) [1].

Among all economic fields, agriculture is one of the most hazardous, with a high incidence of accidents, sometimes even fatal, and responsible for occupational diseases. There are in fact numerous occupational risk for the agricultural workers. These range from traditional ones such as location in disadvantaged, dangerous territories due to slopes and gradients, to the use of equipment and machinery requiring specific training, and also to new and increasingly important risks, which arise from the ever changing multifunctionality associated with agriculture. In addition, farm workers use dangerous and toxic chemical products and they are subjected to hazardous consequences linked to new practices, like farm housing, production of renewable energy, and so on [2].

Unfortunately, the demographic structure does not contribute to improve this scenario. The Italian agricultural system is composed of many small agricultural companies, where almost all the workforce is derived from within the familial context (relatives) of the agricultural business owner, who often is quite old. In fact, in 2017, agricultural companies run by families or by a not economically active owner comprised about two-thirds of the total, while big companies represented the other third. The average agricultural surface of an Italian agricultural company is about 20 hectares, with two employees on average. Over half of companies do not have any employees [1].

According to the data available from INAIL (Italian National Institute for Occupational Accident Insurance), accident’s reports in agriculture have consistently decreased over time, settling at 38.021 in 2015. The data comparing accident’s reports is quite reassuring as a decrease of 3.1% was noted between the years 2014 and 2015. This decrease was even more marked if calculated between 2011 and 2015 where the decrease was of 20%. This trend has continued to decrease in recent years, reaching 33.677 in 2018 [3]. Looking at the data regarding fatal accidents in agriculture is also interesting in that these decreased from 181 in 2014 to 168 in 2015, a decrease of 13 in one year.

Overall, about 80% of the whole reports regard male workers, while for fatal accidents this number decreases dramatically (−18.8%, in three-year period 2014–2018) [3].

Regarding the age class mostly involved in this sector, these injuries are most common in the age group 45 to 54 years. This is in contradistinction to other activities where the age group mostly at risk is usually over 55 years. Thus, it would appear that working in an agricultural environment is more dangerous for a younger cohort of workers [4]. Also, statistics show that 56% of agricultural accidents happen while using mechanical tractors [5].

According to INAIL statistics, 74.1% of occupational diseases in agriculture are musculoskeletal disorders. Among these, farmworkers have reported contusion (30.7%), followed by dislocations and sprains (22%), fractures (21.2%), open wound (20.1%). Anatomical loss was reported in 0.9% percentage of the acute lesions reported (Figure 2) [6].

In 2018, the hands were the most affected body part, which sustained an injury (20.5%), followed by the ankle (11.5%), knee (10.7%), wrist (4.7%), eyes (4.6%), foot (4.3%), and elbow (2.1%). Overall, about one-third of agricultural accidents involved the upper extremity (Figure 3) [6]. As reported also in the international literature, upper limb’s injuries occurring in the farm environment can be very serious, resulting in a relevant loss of working days, often leading to amputation of at least one finger and sometimes to permanent disability [7,8,9].

In this scenario, sustainability in agriculture must be considered as a key factor for its development, with the major climate changes and the challenges of the global market [10,11]. Sustainability in agriculture is related to working processes, which should follow strict rules for safeguarding the environment, for example in the use of pesticides or the disposal of chemical waste, as well as in the maintenance of a consistent production [12]. It also refers to the health and safety of the farmworkers. In fact, any activity causing damage to workers cannot be defined as sustainable. Therefore, the prevention of injuries and work-related disease is a key factor in making agriculture sustainable. Keeping a safe and healthy environment also contributes to improve the overall economic profit of the company, by influencing the delicate balance between profit and management costs [13].

Given these relevant premises, the aim of this systematic review is to investigate the scientific literature to define the characteristics of upper limbs injuries, in particular of the hand, among workers in the agriculture sector while trying to understand the most hazardous activities and the principal causes.

## 2. Materials and Methods

This systematic research follows the Prisma Statement [14].

### 2.1. Literature Research

The research included articles published on the major online databases (PubMed, Cochrane Library and Scopus), up to 30th April 2020. The search strategy used a combination of controlled vocabulary and free text terms based on the following keywords: farmworker, agricultural, accident, injury, upper limb, hand, arm. The search string used combined words such “AGRICULTURE,” “FARM,” “AGRICULTURAL WORKERS,” “FARM EMPLOYER,” with “OCCUPATIONAL INJURY,” “WOUND,” “TRAUMA,” and “UPPER EXTREMITY,” “HAND,” “UPPER LIMB,” and “RISK,” “HAZARD.” All research fields were considered. Additionally, we practiced a manual search on reference lists of the selected articles and reviews so as to carry out a wider analysis.

Two independent reviewers read the titles and abstracts of the reports identified by the search strategy. They selected the relevant reports according to the inclusion and exclusion criteria. Doubtful reports were discussed with a third separate researcher. Subsequently, the researchers independently screened the corresponding full text to decide on final eligibility. Duplicate studies and articles without full texts were eliminated.

### 2.2. Quality Assessment

Three different reviewers assessed the methodological quality of the selected studies with specific rating tools. We used INSA method “International Narrative Systematic Assessment” [15] to judge the quality of narrative reviews, AMSTAR to evaluate systematic reviews [16] and the Newcastle Ottawa Scale to evaluate cross-sectional, cohort studies and case control studies [17]; while the JADAD scale was applied for randomized clinical trials [18].

### 2.3. Eligibility and Inclusion Criteria

The studies included in this review focus on injuries in the agricultural sector, in particular traumatic accidents of the upper limb involving all structures (musculoskeletal, osteotendinous, neurovascular).

### 2.4. Exclusion Criteria

We have excluded publications regarding: non-agricultural jobs, chronic occupational diseases, general injuries if not involving upper limb, studies concerning exclusively children, adolescents and farmers’ family members. Editorial articles, individual contributions and purely descriptive studies published in scientific conferences, without any quantitative and qualitative inferences were also excluded as these were deemed to be less academically robust. The authors did not apply time or linguistic restrictions.

## 3. Results

The online research yielded 951 references: PubMed (715), Scopus (205) e Cochrane Library (31). Of these, 886 were excluded because they were not related to the upper limb accidents in agriculture. Of the remaining 65, 12 papers were eliminated as per the exclusion criteria.

Of the 53 studies that were included in this systematic review (Figure 4), 3 are narrative reviews and 50 are original studies. A total of 35 of the original studies are cross-sectional studies, 11 case series, 2 case reports, 1 cohort study, and 1 case-control study; no systematic reviews were found (Table 1).

Most of the published studies used in this review (24 articles) originated in the United States. Five papers were published in 2013 and another 4 papers published in 2015 (Table 1).

### 3.1. Reviews

Regarding the narrative reviews (Table 2), the INSA score shows an average of 6.5, median and modal values of 4, thus indicating an intermediate quality of the studies. The most appropriate methodological review was conducted in United Kingdom (INSA = 5).

Three narrative reviews, which have analyzed upper limbs injuries in the agriculture sector (3/49; 6.1%) were also identified. According to two out of the three reviews, upper limbs injuries are the most common agricultural accidents requiring access to a hospital and may represent just under half of the whole injuries in an agricultural setting [19,20]. All the three papers state that the largest part of the victims were males and the median age was between 39 and 44; generally, younger workers seem to be more at risk for acute injuries in the agriculture sector [21], but upper limbs injuries may happen even in older people [19,20]. These injuries tend to have a seasonal pattern, occurring mostly during the months associated with harvesting [19,20].

All reviews state that the most frequent injuries manifest open wounds, with a wide range of severity. Angoules, 2007 [19] reports that, in half the cases, the amputation of at least one finger occurs and time for returning to work is at least 25 days on average [19,20]. Agricultural machinery was the first cause of injuries of the upper limbs in farms, followed by animal-related injuries [19,20]. The main machines causing these injuries are tractors, power off devices, augers, hay balers, combine harvesters, and corn pickers; each of these causes a slightly different pattern of lesions between one another [20]. Regarding the immediate medical management of these injuries, two papers point out that there is a high risk of contamination and subsequent infection, so that tetanus and antibiotic prophylaxis is recommended; wound examination, irrigation, and surgical debridement are performed in most cases [19,20].

### 3.2. Cross Articles

The scores assigned to the original papers have an average value of 5.42, a median and a modal of 6 (Table 3). This situation amounts to an intermediate quality of the studies; an American study obtained the highest values (New Castle = 9).

#### 3.2.1. Part of the Body Injured

Among the 35 cross sectional studies, five studies focus exclusively on hand lesions (14.2%). The other 30 refer to lesions of the general upper extremities (85.7%); among these, 16 studies gave some references about what part of the upper limb was involved while 14 cross sectional studies describe in their sample many lesions of the hand. 

Ten of the 35 studies (28.5%) specify some kind of injury to the fingers: seven studies claim that fingers are the most effected part of the upper limb. For example, in Mehri, 2017’ [22], about two-thirds of the cutting injuries to fingers occurred at work. 21.5% of the subjects had experienced similar accidents before and in two-third of cases, the accident occurred in the fingers of their dominant hand. In addition, Pate, 2013 has found a greater risk of fingernail loss in individuals working at non-certified farms [23]. 

#### 3.2.2. Type of Lesion

33/35 (94.2%) of the studies characterized the reported injuries. In 28 studies (80%), the most frequent type of lesion was an open wound, starting from bruising, laceration, and cutting lesions up to mangling lesions. In other studies, the authors have reported other mechanisms, such as fractures, strains, sprains, and overexertion lesions.

Among the studies characterizing injuries, 50% include primary amputation of some part of the upper limb due to agricultural accidents in their records.

In 12 studies (34.2%), there is a percentage of injuries involving long-term disability and/or amputation. According to Cogbill, 1991 [24] corn picker injuries, power take-off entanglement, and farm machinery accidents in general were more likely to result in disabling conditions.

Sometimes, amputations result from superadded infections. For example, Ali’ study, 2008 [25] reports that, of 214 upper limb open wounds caused by agricultural accidents, 40 developed an infection after the injury. Of these, 17 were superficial, 16 involved infection of the deep soft tissue, seven developed osteomyelitis, and six went on to amputation due to the infection. In addition, Obradovic-Tomasev 2016 [26] have found that 15% of his sample developed a fungal infection principally because of Aspergillus in hand injuries caused by a corn picker, Ozyurekoglu 2007 [27] found that all patients with severe muscle, tendon, and nerve injuries at the forearm or elbow level developed an infection, with one case requiring revision amputation at the mid-forearm level.

Pate’ tests (2013) [23] showed a significant presence of microbiological pathogens, in particular of Salmonella and Staphylococcus, in minor open wounds of workers employed in farms, which had not invested in a US government certification regarding hygiene and preventive measures at work. 

Three studies report that irrigation and surgical debridement are needed for most of the cases [28,29,30]. In Grandizio’s study 2018 [28], patients with upper extremities injuries are more likely to require surgery and to be readmitted to hospital than those with other kind of lesions, especially with some risk factors for readmission such as age > 18 years, falls from height and surgery.

#### 3.2.3. Causes of Injury

Most of studies describe the causes of accidents (32/35; 91.4%). Twenty studies (57.1%) report that the most frequent cause of injury is an accident with an agricultural machine, mainly caused by an overturning tractor but also with all the other types of machines present on farms, such as grain augers, power take-off, hay balers, corn pickers, and wheat thresher. According to Cogbill 1991 [24], the most frequent machines involved in upper limb accidents are power take off and corn pickers. 

On the other hand seven studies (20%) reported that hand tools, like sickle, axe, spade, handsaw, and hoes were the principle culprits; these studies are from developing countries, such as Nepal and India, where agriculture is not fully mechanized as yet.

Another cause cited by seven authors (20%) is injury resulting from direct contact with animals. For example, in Swamberg’ study 2013 [31], horses are more frequently the direct source of injuries compared to non-horse related events, comprising 56.8% of the documented injuries. In this case, the upper extremities accounted for the highest number of injuries, mainly the wrists, fingers, hands (17.8%), arm, and shoulders (17.0%). Lindsay 2004 [32] reports that clipping injuries affect the upper limbs more than other livestock activities and he cites some risk factors, among which are working alone, working with beef cattle and a younger age.

Some authors have highlighted other risk factors that lead to greater fatigue and distractions for farmworkers. For example, in the study of Bhattarai 2016 [33], a working experience of over 20 years negatively correlated with frequency of injuries, while having worked for >48 h raised the risk of injuries. Layde 1995 [34] reports that the number of hours worked per week is significantly associated with injuries, in fact there is a 2% increase in rate per additional hour worked. For Ravi 2019 [35], a large proportion of injuries (55; 60.4%) occur after working between 8 and 12 h. In fact, working hours of the patients at the time of injury were about 6.50 ± 1.14; twenty-three (44%) patients admitted that they had experienced fatigue/over-work at the time of injury and thirteen (25%) patients had made use of alcohol/drug to overcome this fatigue.

Khurram 2015 [30] has reported that the most common pre-disposing factors for injuries were lapses in concentration while operating machinery, such as loss of synchronization between the person moving the roller and feeding the crop. Kogler 2016 [36] reports that the loss of machine control was a main cause of accident. According to Pickett 2001 [37], three quarters of the injuries reported were found to be either due to carelessness or else no specific reason was identified. Other factors that victims think may have contributed to accidents, included tiredness, inexperience, and lack of safeguarding on farm machinery. Mehri 2017 [22] found that 10% of the victims were injured while responding to a mobile phone call, 3% of the victims reported that they were listening to music through headphones before the accident, 5% reported a quarrel with a colleague, and 11.5% reported a quarrel with their employer prior to the accidents.

Finally, some authors highlight the preventive role of safety devices. For Rabbani 2018 [38], only 1% and 3% of the participants used gloves and masks, respectively, and none of the participants reported the use of long boots. In Ravi 2019’s article [35], four injuries could have been prevented by using footwear as a form of personal protective equipment. In the study of Bhattarai 2016 [33], one-third of the farmers report that they use personal protective equipment; however, almost all of these workers only use ordinary masks at work.

#### 3.2.4. Time of Injury

Nineteen studies (54.2%) have described in which part of the year the injuries occurred. For 12 studies (34.2%), injuries were more frequent in summer, in particular, between June and August; on the other hand, in six studies, injuries were frequent even in autumn, in particular from October to December. In Bhattarai 2016 [33]’s research, injuries occurred more often during the rainy season. Only one study, regarding animal farms, showed that winter is the most common season for injuries. According to Pickett 2001 [37], during the winter, accidents involving animals are proportionally more frequent and in the study of Watts 2011 [39] the peak of cattle-related injuries was in September, in concomitance with the beginning of calving and milking of cows.

Regarding the work shift, the data are somewhat controversial. Six articles (17.1%) report more injuries in the afternoon, in particular around 5 pm, while for Athanasiov 2006 [40], 42% of accident cases occurred at the beginning of the work shift, 35% in the middle and 23% in the end of work shift. Also, Bhattarai 2016 [33] sustained that two-thirds of lesions happened during the morning shift.

#### 3.2.5. Demographic Characteristics of Injured Workers

Most of the studies specify the gender of the injured workers (32/35; 91.4%). In 30 studies (85.7%), the prevalence of injuries involved male workers. Only two studies, one from Nepal and one from India, show the same incidence of agricultural injuries between males and females.

Thirty-four studies (97.1%) report on the age of the patients: 13 studies (37.1%) report the mean age of the victims, while 21 studies (60%) report the age groups. Among the first, the age group mostly exposed are between 35 and 43 years of age.

Only three studies (8.5%) report that workers aged 55 and over were involved. In Alexe 2003’s publication [41], the age group which sustained most injures was that between 55 and 64; in Hansen 1999‘s article [42], this was 45–64 and finally, in Hartling 1998’s [43] study, there is a notable incidence of accidents in people over the age of 64 years. In addition, Alexe 2003 [41] reported a difference in age between locals and migrants involved: in the former, injuries in the upper limb occur more often in the age group between 55–64 years, while in the latter this occurred in the age group between 15–34 years.

The study of Allen 2015 [44] reports that upper limb fractures are equally distributed between genders and age groups, while upper limb open wounds are more frequent in males but with no difference between age groups.

Some researchers have identified a higher accident risk in low educational status. For example, Patel 2018 [45] found that the rate of accidents among illiterate workers was 59.9% (133 cases) compared to below and above matriculation level; for Wang 2011 [46], workers with less than 12 years of education accounted for 79.2% of the injuries identified in his study. Also, in the study of Khurram 2015 [30], most of the workers were semi-literate or illiterate and belonged to lower or middle socioeconomic status, with a total disregard for safety regulations.

### 3.3. Case Series and Case Report

Among 13 articles, 12 studies describe injuries caused by farm machinery and only an article reports an accident occurring by a farmyard gate (Table 2). Often farmers were injured when the machine appeared to have stopped only to resume functioning when it became unclogged. Three case series studies report specific injuries of the hand due to corn picker accidents, while one author reported upper limb lesions due to threshing machines. Six case reports involved hay balers, 1 combine harvester, 2 power take-off, and 2 auger flighting. All the subjects were male, within a wide age group, ranging from 17 to 71 years.

The type of lesions described were all open wounds, with a different grade of damage of the soft tissue and bones. Salibi 2017 [47] reported a case of mucormycosis caused by a degloving injury of the arm which occurred in the farmyard. Almost all the cases ended up with an amputation, ranging from just the phalanx of one finger to an upper limb amputation. In almost all cases some kind of surgical intervention was performed.

Corn picker injuries happened mostly in corn picking season that is from September to December [48], the threshing machines injuries occur more often in August [49] and hay balers’ accidents occur principally in June and October [27]. The 2 auger flighting accidents occurred both in September [50]. Also, for Copuroglu 2012 [51], the highest number of accidents was in summer.

## 4. Discussion

About 1.3 billion of people are employed in agricultural sector worldwide [52]. Although in Italy agricultural accidents have constantly decreased, injuries in this sector still represent an important burden of disease, often leading to physical impairment and even to death. In fact, about one quarter of the injuries resulted in some disability in the last years [6]; disabling injuries cause not only a relevant impairment for the victims, who sometimes cannot go back to their usual job, but at the same time it creates a social problem with a burden on health care system and also on society in general [53,54,55].

In this review, the studies selected are focused mostly on upper limb injuries. These lesions in agriculture sector can be extremely severe, often leading to amputation, either primarily or in consequence of surgery. These data are quite discordant to national Italian data where only 0.9% of all agricultural related injuries end with anatomical loss. This can be explained by the various geographical provenience of the studies reported in this review as well as by the fact that the majority of studies are conducted on hospital patients and probably only the most severe injuries are reported.

Farm machinery use could explain these lesions; in fact, machines are indeed recognized as the main source of accidents in agricultural industries [56,57,58].

Also in our review the machine most involved in accidents was the tractor. This finding tallies with national data which report that tractors are involved in farm’s accidents half of the time [6]; in addition, international literature highlights that 4–14% of non-fatal injuries [59,60,61] and over one-third of fatal injuries involve this vehicle [62]. Several other machines are involved in upper extremities injuries such as grain augers, hay balers, power take-off, wheat threshers, and corn pickers. Often these injuries caused by these machines are mutilating and represent a real emergency, requiring major surgery [30,63,64,65].

Even animals represent a notable source of injuries, but less than machinery-related accidents. This contrasts with other studies in agriculture [66,67,68].

In developing countries, the most frequent injuries suffered in the upper limbs are due to hand tools like axe, spade, sickle, hoe, and others because agriculture is still not mechanized and machines are not used much by farmers [33,69,70]. This aspect influences the severity of wounds, which are usually minor. Moreover, these studies are conducted through surveys administered to farmers resulting in a more precise report of injuries, even of the trivial ones.

Mechanization has certainly made the farm work more sustainable for agricultural workers reducing heavy workloads in many strenuous tasks but at the same time, lack of safety mechanism, high noise, pollution and vibration can turn machines in a major source of dangers within the farm environment. Mechanization can reduce heavy workloads, especially those related to musculoskeletal disorders. Attention to ergonomic intervention, farming equipment modification and system design is needed to improve working conditions, together with environmental protection, social and economic stability [71,72].

The most frequent type of lesion reported for the upper limb is an open wound, which results from a high-energy trauma involving soft tissues and sometimes even the bones, with an increased risk of infection and healing complications [19,20,25,51]. The wounds are often grossly contaminated and some of the in-site bacteria can be antibiotic resistant [25]; infection complicates the healing of the injury and, in the most serious cases, can lead to osteomyelitis with secondary amputation. This makes prophylactic antibiotic administration crucial for the correct management of wounds [19,20,51].

Regarding the demographic characteristics of the victims, we have found that men are more at risk than women [73,74]; this report is in accordance with the literature, as well as with INAIL’s Italian national data. Probably, men are exposed more because they are more employed in agricultural industry, in more strenuous tasks [37,75].

In contrast to other kind of industries, in agricultural activities, younger people have a higher risk of becoming victims of farm injuries [76,77,78,79,80,81]. According to our findings, upper limb injuries confirm this trend; in fact, the most affected age group in this review is between 20 and 45 years and sometimes victims are even younger [82,83]. This is partially explained by a young employed workforce, in developing and in high-income countries, such as Italy; most countries employ young seasonal migrant workers during the busiest periods of the year [84,85]. However, in Italy, the most hit age class is 45–54 year old, probably because of the overall older working population in our country.

In this review, the periods of the year with a higher rate of accidents are summer and the beginning of fall. From June to November, the workload is massive, characterized by long shift hours and a few rest days. The huge engagement can possibly explain the high rate of injuries in these periods; farmers are urged to complete a large amount of work in the shortest time possible and in situations that may not be fully compliant with the best safety regulations [29]. On the other hand, animal-related accidents of the upper limbs seem to register a higher incidence of injuries during the winter [37,40].

According to our findings, we can state that fatigue can be a contributory cause of upper limb farm injuries [51,86,87,88]. Often victims reported that the main cause of the injury was carelessness, inaccuracy, and distraction [37,48,76,87,89]. 

This systematic review has some limitations. First, in our analysis, we have not found any trials that are fundamental for understanding the determinants of occupational diseases and founding new appropriate interventions.

Studies are often conducted outside the European context, in small and selected groups of workers. Overall, the quality of the studies was not too high, because of the frequent use of subjective assessment tools, such as questionnaires not always being standardized, poor description of the sample characteristics and of the subjects that were possibly excluded.

In addition, it is important to point out the source of information used to conduct the studies. Principally, the major source of information for farm injuries are three: hospital records, surveillance systems or insurance records, and interviews administered to farmers. Injuries reported from these sources can be very heterogeneous, especially in terms of gravity and type of lesions; reports from hospital tend to consider only serious cases and therefore show an epidemiology characterized by few severe accidents. On the other hand, surveys show a more realistic dimension but they may lack accuracy, in description and gravity of the injury. Finally, through insurance records or surveillance system, we can find misreporting of number of injuries, considering that relevant number of farm workers are not registered, especially among seasonal employers and migrants.

## 5. Conclusions

The agricultural sector employs an estimated 1.3 billion workers worldwide, that is half of the world’s labor force. In terms of fatalities, injuries, and work-related ill-health, it is one of the three most hazardous sectors of activity, because of the particularly hazardous nature of agricultural work as also recognized by ILO in its Safety and Health in Agriculture Convention (No. 184), adopted in 2001. Agricultural workers deal with domestic and wild animals, work in a wide range of climatic environments and often their farm is small and runs under a family management. Upper limb injuries represent an important burden of disease among agricultural employees. The upper extremities and the hands are common sites of injuries: accidents can be serious, often requiring surgery and a long recovery period. The real impact of these injuries on the health of the workers and on public health is still partially unknown and more research is needed to identify the main risks and effective preventive strategies.

A health promotion system and good practices are therefore needed to support farms, especially the small ones, to make them grow in terms of prevention strategies and safety management. In this whole context, specific programs are required to support agricultural companies and help them correctly follow the most recent national and regional regulations, in terms of the safety of the equipment and products, as well as in education and training of workers. The awareness of the sources of risks sources and the understanding of dangerous situation in farms set the basis for sustainable agricultural work and for the management of innovative organizational behavioral measures aimed at reducing work related possible health effects.

## Figures and Tables

**Figure 1 ijerph-17-04501-f001:**
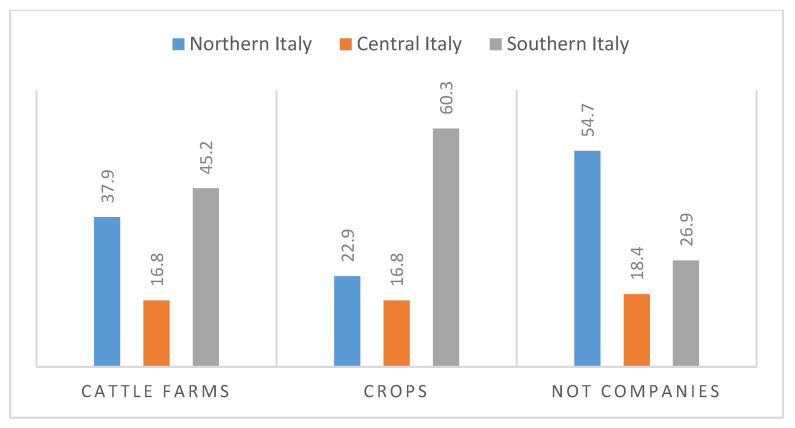
Distribution of farms in Italy, percentage values [1].

**Figure 2 ijerph-17-04501-f002:**
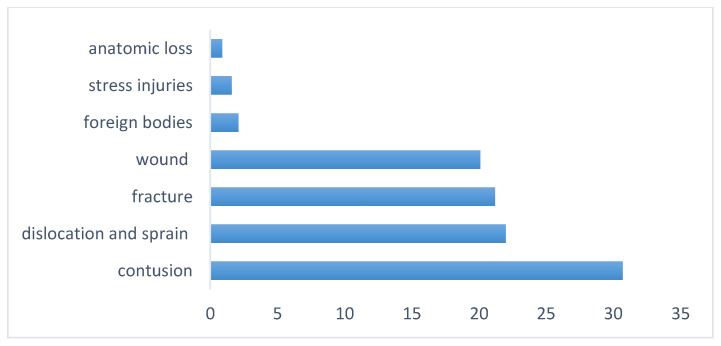
More frequent injuries in agriculture, percentage values [4].

**Figure 3 ijerph-17-04501-f003:**
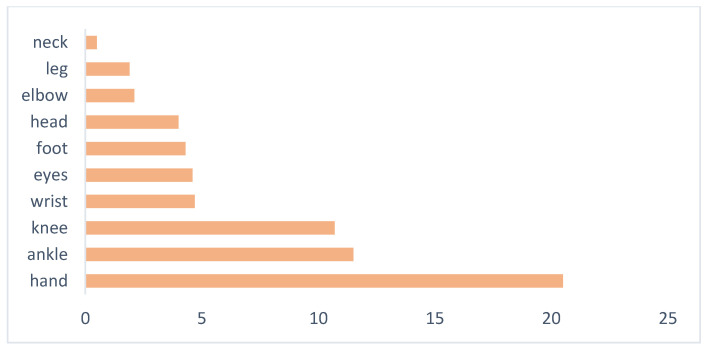
More involved part of body in agriculture, percentage values [4].

**Figure 4 ijerph-17-04501-f004:**
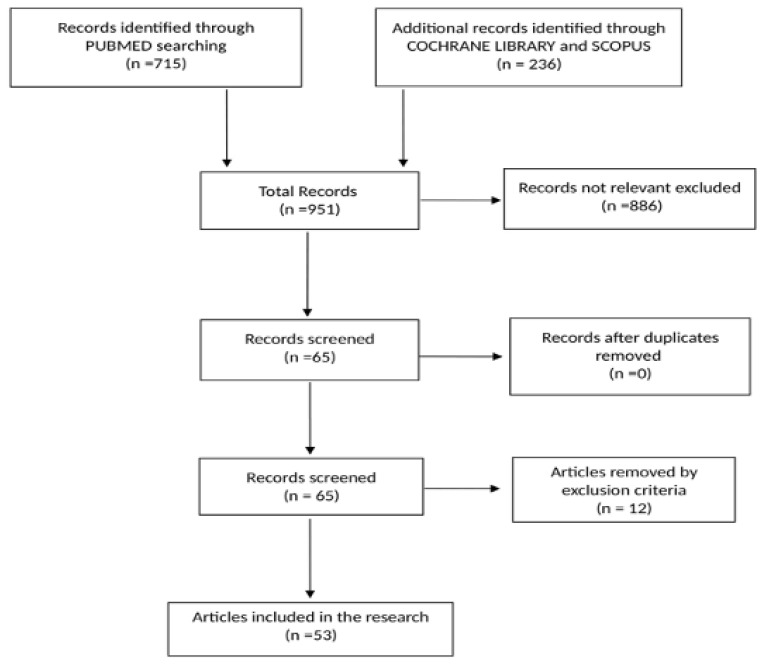
Bibliographic research flow chart.

**Table 1 ijerph-17-04501-t001:** Included studies in systematic review.

First Author	Year	Type of Study	Country	Subjects
**Alexe D.M.**	2003	Cross sectional	Greece	N.4326
**Ali Mir H.**	2008	Cross sectional	USA	N.216
**Allen D.L.**	2015	Cross sectional	USA	N.2295
**Angoules A.G.**	2007	Narrative review	UK	
**Athanasiov A.**	2006	Cross sectional	Australia	N.52
**Bhattarai D.**	2016	Cross sectional	Nepal	N.500
**Campbell D.C.**	1979	Case series	USA	N.51
**Cogbill T.H.**	1991	Cross sectional	USA	N.739
**Copuroglu C.**	2012	Case series	Turkey	N.41
**Das B.**	2013	Cross sectional	India	N.124
**DavasAksan A.**	2012	Cross sectional	Turkey	N.5027
**Davis K.G.**	2007	Narrative review	USA	
**Ennaciri B.**	2015	Case report	Morocco	N.1
**Gainor B.J.**	1983	Case series	USA	N.3
**Gorsche T.S.**	1988	Case series	USA	N.15
**Grandizio L.C.**	2018	Cross sectional	USA	N.273
**Hansen R.H.**	1986	Cross sectional	USA	N.64
**Hansen T.B.**	1999	Cross sectional	Denmark	N.260
**Hardin C.A.**	1967	Case series	USA	N.2
**Hartling L.**	1998	Cross sectional	Canada	N.1023
**Ingram M.W.**	2003	Case series	Canada	N.4
**Isik D.**	2012	Case series	Turkey	N.25
**Jawa R.S.**	2013	Cross sectional	USA	N.106
**Kalenak A.**	1978	Case series	USA	N.7
**Khurram M.F.**	2015	Cross sectional	India	N.161
**Kogler R.**	2015	Cross sectional	Austria	N.84
**Kogler R.**	2016	Cross sectional	Austria	N.3250
**Layde P.M.**	1995	Case Control	USA	N.311
**Lindasy S.**	2004	Cross-sectional	UK	N.591
**Marais S.**	2005	Cross sectional	South Africa	N.500
**Mc Curdy S.A.**	2004	Cross sectional	USA	N.136
**McKinnon D.A.**	1967	Case series	USA	N.12
**Mehri N.**	2017	Cross-sectional	Iran	N.200
**Melvin P. M.**	1972	Case series	USA	N.30
**Obradović-Tomašev M.**	2016	Cross sectional	Serbia	N.60
**Özyürekoğlu T.**	2007	Case series	USA	N.21
**Pate M.L.**	2013	Cross sectional	USA	N.85
**Patel T.**	2017	Cross sectional	India	N.50614
**Pfortmueller C.**	2013	Cross sectional	Switzerland	N.815
**Pickett W.**	1995	Cross sectional	Canada	N.1364
**Pickett W.**	2001	Cross sectional	Canada	N.8263
**Pratt D.S.**	1992	Cohort study	USA	N.600
**Rabbani U.**	2018	Cross sectional	Iran	N.472
**Ravi S.**	2019	Cross sectional	India	N.400
**Salibi A.**	2017	Case report	UK	N.1
**Schwab C.V.**	2000	Cross sectional	USA	N.437
**Singh R.**	2005	Cross sectional	India	N.52
**Swanberg J.E.**	2013	Cross sectional	USA	N.284
**Wang S.**	2011	Cross sectional	USA	N.13604
**Watts M.**	2011	Cross sectional	New Zealand	N.78
**Yaffe M. A.**	2014	Narrative review	USA	
**Young S.K.**	1995	Cross sectional	Canada	N.72
**Zhou C.**	1994	Cross sectional	USA	N.1000

**Table 2 ijerph-17-04501-t002:** Reviews, case series, case report, control, and cohort studies with relative scores.

Author	Study	Injury’s Characteristics	Cause	Involved Part of the Body	Score
**Angoules A.G.**	Narrative review	Open wounds, lacerations, amputation	Machinery (tractors, corn pickers, conveyor blets, mowers)	Upper limb	I.5
**Davis K.G.**	Narrative review	Lacerations, punctures	Agricultural machinery	Hand	I.4
**Yaffe M. A.**	Narrative review	Bone and soft tissue injuries, amputations	Various machinery (tractors, augers, hay balers.)	Upper limb, hand	I.4
**Campbell**	Case series	Skin avulsion, laceration, fractures, amputation	Corn picker	Hand	n.a.
**Copuroglu C.**	Case series	Open and close fractures	High energy injuries; farm machinery, tractors	Hand, forearm, elbow, upper limb	n.a.
**Ennaciri B.**	Case report	Deep wound	Blades of a combine harvester	Hand	n.a.
**Gainor B.J.**	Case series	Wound, open fracture, amputation, burning	Hay baler	Forearm, hand	n.a.
**Gorsche T.S.**	Case series	Mutilating injuries	Corn picker	Hand, hand’s fingers	n.a.
**Hardin C.A.**	Case series	Mangling injuries, amputation	Hay Baler	Upper limb	n.a.
**Ingram M.W.**	Case series	Amputation	Auger flighting	Thumb, fingers, hand	n.a.
**Isik D.**	Case series	Soft tissue injury, tendon injury, open fractures, amputation	Threshing machine	Fingers, hand, forearm	n.a.
**Kalenak A.**	Case series	Amputation, open fracture, avulsion injury	Tractor power take-off, corn husking power take-off	Arm, hand	n.a.
**Layde P.M.**	Case Control	Lacerations, contusions, fractures, amputation, avulsion, burn	Agricultural machinery	Hand, arm, wrist, shoulder	N.5
**McKinnon D.A.**	Case series	Amputation, mangling injuries, burns, skin avulsion	Hay Baler	Upper limb	n.a.
**Melvin P. M.**	Case series	Open wounds, open fractures, crushes, burns	Corn picker	Hand, fingers	n.a.
**Özyürekoğlu T.**	Case series	Amputation, open fractures, lacerations, burns	Hay baler	Upper limb	n.a.
**Pratt**	Cohort study	Not specified	Machinery, animals, falls	Hand, arm	N.6
**Salibi A.**	Case report	Degloving injury and fracture	Crush in farmyard gate	Hand, distal forearm	n.a.

**Table 3 ijerph-17-04501-t003:** Cross articles with relative scores.

First Author	Injury’s Characteristics	Cause	Involved Part of Body	Score
**Alexe D.M.**	Open wounds, contusion, concussion, fracture	Instruments, animals, machinery	Upper limb	N.7
**Ali Mir H.**	Open wounds, laceration, fractures, amputation	Not known	Hand, fingers	N.6
**Allen D.L.**	Fractures and open wounds	Not known	Upper limb	N.6
**Athanasiov A.**	Not specified	Augers, especially contact with augers flighting	Fingers, hand, arm	N.5
**Bhattarai D.**	Cuts, punctures, laceration, fracture	Hand tools, slipping, animals, sharp instruments, fall	Finger, hand	N.5
**Cogbill T.H.**	Not specified, lacerations, fractures, amputations	Farm machinery, corn picker, power take-off, farm animals, tractor, fall	Upper extremity	N.5
**Das B.**	Mostly cut injuries and laceration	Hand tools, machinery (tractors, animal drawn puddlers, motor sets)	Hand fingers, hand, wrist	N.4
**DavasAksan A.**	Traumatic amputation, open wounds, fractures	Agricultural machinery	Hand, wrist	N.6
**Grandizio L.C.**	Fracture, upper extremity mangling injury	Animal, falls, table saw, machine, farm vehicle	Upper extremity, hand, fingers	N.6
**Hansen R.H.**	Open wounds, fractures, amputations	Tractors, augers, power take-off, combines, plow, corn sheller	Upper extremity	N.5
**Hansen T.B.**	Lacerations, amputations, fractures	Machinery (corn and potato pickers), animals (fractures)	Hand and fingers	N.5
**Hartling L.**	Open wound, fractures, traumatic amputation	Entanglement in machinery	Upper limb, fingers	N.5
**Jawa R.S.**	Laceration, amputation, crush injuries	Machinery (tractor, auger, baler, picker, grinder, feeder)	Hand, wrist, upper extremity	N.6
**Khurram M.F.**	Amputation	Wheat thresher, fodder cutting	Hand, fingers	N.5
**Kogler R.**	Wounds, fractures, superficial injuries	Mowing machines	Upper extremity	N.6
**Kogler R.**	Wounds, fractures, dislocations, amputations	Agricultural machinery	Upper extremity	N.6
**Lindasy S.**	Bruising, lacerations, fracture	Taggling and clipping cattle	Upper limb	N.5
**Marais S.**	Cuts, bruises, abrasions	Working in the orchards (not more specified), motor vehicle	Hands, fingers, shoulder	N.5
**Mc Curdy S.A.**	Open wounds, sprains, strains, fractures, amputation	Overexertion, machinery, falls, animals, cutting instruments	Fingers, hand, forearm, upper arm, elbow	N.9
**Mehri N.**	Amputation	Hay maker, machinery, cutting tools, heavy object dropping	Hand, fingers	N.6
**Obradović-Tomašev M.**	Mutilating and destructive injuries, skin tear	Corn picker	Hand	N.4
**Pate M.L.**	Cuts, abrasions, puncture, fingernails loss	Not specified	Hands, fingers	N.5
**Patel T.**	Non-fatal injuries, not further specified	Hand tools (spade, axe, sickle), falls, animals	Han, wrist, forearm, elbow, shoulder	N.6
**Pfortmueller C.**	Not specified	Motor vehicle, cutting machine, tools, saw	Upper limb	N.6
**Pickett W.**	Fractures, open wounds	Machinery (tractors, grain augers, power take-off, balers), falls, animals	Upper limb	N.6
**Pickett W.**	Sprains, strains, fractures, lacerations	Machinery, lifting, animals,	Upper limb	N.6
**Rabbani U.**	Cuts, fracture	Hand tools (garden hoe, harrow, shovel, sickle), animal handling	Hand, upper limb	N.6
**Ravi S.**	Lacerations, abrasions	Plant thorns	Hand, fingers, lower limbs	N.5
**Schwab C.V.**	Laceration, fractures, amputation, mangling injury	Auger	Hand, fingers, arm	N.6
**Singh R.**	Not specified, amputations	Wheat thresher	Hand/wrist, hand/forearm	N.4
**Swanberg J.E.**	Strain, contusion, fracture, cuts, sting	Horses, tools, lifting, falls, insects/plants	Wrist, fingers, hand, shoulder	N.6
**Wang S.**	Lacerations, strains, fractures	Contact with objects, falls, bodily reactions	Hand, wrist, fingers	N.5
**Watts M.**	Contusions, lacerations, sprains, abrasions, fracture	Handling cattle (kicks, crushes, head butts)	Hand, wrist, elbow, forearm, shoulder	N.5
**Young S.K.**	Fractures, amputations, crush injuries, lacerations	Machinery, animals	Upper limb	N.5
**Zhou C.**	Sprain, contusions, fractures	Tractors, falls, animals	Hand, wrist, fingers, legs	N.6

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
