# Peer review of "Upper Limb’s Injuries in Agriculture: A Systematic Review"

_ijerph, 2020, doi:10.3390/ijerph17124501_

Round 1

Reviewer 1 Report

The paper proposes a literature review analysis about specific type of injuries occurred in the agriculture sector. Differently from traditional papers about injury analysis, the review is based on a paper study based assessment. Although the review process is well detailed, the more attention must be paied in outlining how some data have been extracted by each paper.

It could be interesting to compare systematically all results obtained with statistical data usually available at national level: some citations about Italian data provided by inail are reported in the text but they are very limited.

Furthermore, some variables need to be described well: what are the main dimensions of the firms deducted by the paper review?what are the main causes evaluated for the injury? The general validity of these results need to be clarified with quantitative data.

Author Response

Dear Reviewer,

we have tried to improve discussions by putting some comparisons with national data, where possible. Your requests are highlight in purple color.

Staying available,

Best regards

Reviewer 2 Report

First of all, this article is important not only for the scientific field occupational health but also to correlated domains such as several areas of engineering like production, agriculture, mechanical, systems, safety engineering, etc., besides human factors and ergonomics, industrial design, economics and political studies.

In order to make this study more robust and consistent for the readers it is recommendable some improvements in terms of content including method and format as listed:

1. Introduction: some bar graphs could be inserted in the text, if there is no other limitation, to facilitate the understanding for the reader when reading some statistics along the text. Some examples could be:

-in the paragraph starting in line 34 a bar graph could inserted from the references giving a better idea about the farm property in Italy;

-in the lines 39 and 40 there is a claim that can supported by a graph compairing with other economics sectors, for example;

-in the the paragraph form line 46 to 52 arguments could be supported by a graph of the cited reference;

-in the text from line 53 to 61 could be supported by a graph as well

-the same approach could be taken for the arguments from line 62 to 75;

2. Method:

- the introduction of this section could be more extensive and maybe include an scheme of the methodological phases;

-in the topic 2.1 Literature review: it is necessary to clarify the search period and also, if used, the query strings making explicit the word combination if the there is any Boolean searches;

-in topic 2.3 Eligibility and inclusion criteria: the following sentence (line 110) must be rewrite to be more clear for the reader and avoid confusion: "No temporal and linguistic restrictions were applied";

3. Results:

-it might interesting for the reader to make some comments why there are no works (articles, reports, etc.) for some top agricuture producer countries like Brazil, Russia, Mexico, Argentina for example or very few related to the biggest agriculture producer in the world, China.

-in the section 3.2 it is recommendable do not start sentences by numbers like what happens in the lines: 167, 173, 239, and 255. Also in this section some paragraphs could be merged for better formatting and reading fluidity.

-in the line 199 and 200 maybe rewrite the sentence informing that the accidents happen during the operation of the agriculture machines, improving the quality of the text. 

-in the section 3.3. the same as pointed out above, to rewrite the sentence adding in the line 274   "...injuries caused by (the operation of the) farm machinery... " to be more precise.

4. Discussion:

- in the line 317 there is a claim which should be mitigate when talking about manual work in developing countries. Some developing countries have the double features: agriculture highly mechanized and manual for example in Latin America countries. 

5. Conclusions:

-to add value to this very good research it might be interesting to add 1 or 2 paragraphs in the conclusion indicating some future and promising research directions in the domain.

Author Response

Dear Reviewer,

we have made the required corrections (yours are highlight in Yellow); we put graphs in introduction and improve the conclusion. In addition, we have corrected results and searched to specify cause of injury, where possible.

Staying available,

Best regards.

Reviewer 3 Report

Some revision of  various words  and sentences  used in the paper need to be made to clarify and improve meaning.

I believe that the means of referencing  some studies by name only  should be up-graded with  date/ number for the benefit of the reader.

I consider that the conclusions section is  too general and it does not relate to the  study finding. I consider that this section of the paper should be revised.

I have attached the paper with comments attached.

Author Response

Dear Reviewer,

we have made the required corrections (yours are highlight in Green). In particular, we have added years in bybliographic references and improved conclusions.

Staying available,

Best regards.

Round 2

Reviewer 1 Report

The paper has been fully improved in several parts. It could be published as is.